# Porous Organosilica Films: Is It Possible to Enhance Hydrophobicity While Maintaining Elastic Stiffness?

**DOI:** 10.3390/polym17172433

**Published:** 2025-09-08

**Authors:** Alexey S. Vishnevskiy, Dmitry A. Vorotyntsev, Dmitry S. Seregin, Konstantin A. Vorotilov, Alexander S. Sigov

**Affiliations:** Research and Educational Center “Technological Center”, MIREA—Russian Technological University (RTU MIREA), 78 Vernadsky Ave., Moscow 119454, Russia; vorotyncev@mirea.ru (D.A.V.); d_seregin@mirea.ru (D.S.S.); vorotilov@mirea.ru (K.A.V.); assigov@yandex.ru (A.S.S.)

**Keywords:** sol–gel, organosilica films, terminal dimethyl, diethoxydimethylsilane, DEDMS, porous films, water contact angle, Young’s modulus, adsorption ellipsometric porosimetry

## Abstract

Organosilica films, composed of a silicon oxide network with terminal methyl groups, are widely utilized in various applications, including microelectronics. Many of these applications require high hydrophobicity and good mechanical properties, which pose a significant challenge because the Si–CH_3_ groups disrupt the Si–O–Si network. This issue becomes particularly pronounced in porous films. Here, we investigate whether material properties can be tuned by simply altering the spatial arrangement of methyl groups. To achieve this, we prepared copolymer films with one or two methyl groups bonded to a silicon atom, while maintaining a constant total amount of methyl groups. The films were deposited using a sol–gel technique combined with template self-assembly. The precursor content was varied to compare films with different proportions of Si–CH_3_ and Si(–CH_3_)_2_. Film characterization included FTIR, ellipsometric porosimetry, AFM, and WCA measurements and dielectric constant evaluations. Our findings indicate that precursors containing dimethyl groups enhance the connectivity of the Si–O–Si network, resulting in a higher Young’s modulus and smaller pore size compared to films with an equivalent amount of methyl groups. However, the lower thermal stability of dimethyl bonds limits the thermal budget of these films. Thus, the spatial arrangement of organic groups within the polymer structure can be employed to tune material properties. These results expand the understanding of organic–inorganic hybrid materials and offer novel approaches for their applications.

## 1. Introduction

Organic–inorganic hybrid films are widely used in many fields, from biomedicine to electronics, as they combine the advantageous properties of both inorganic and organic materials [1,2,3,4]. Sol–gel chemistry offers a unique approach to obtaining organic–inorganic hybrids with different compositions by using several precursors that react with each other to form a metal oxide network and link organic chains [5]. One of the best-known and most important materials for various applications is organically modified silicates (ORMOSILs) or organosilica glasses (OSG) [6,7,8,9]. These materials, based on a silicon–oxygen network, offer good thermal and chemical stability, adhesion, mechanical, optical, and electrical properties. Their good compatibility with materials and processes used in microelectronics allows for their use in semiconductor manufacturing [10].

One of the most important microelectronic applications of silicon dioxide is as an insulator in the interconnections of integrated circuits (ICs). Silicon dioxide has been used for this purpose for many decades due to its excellent dielectric properties. Continuous advancements in speed and reductions in the size of microelectronic devices are driven by the introduction of innovative technologies and materials with enhanced properties. However, as the density of components increases, issues related to crosstalk in transistor structures emerge. Addressing this challenge requires a comprehensive approach, particularly the implementation of new conductors with lower resistance and insulating layers with reduced dielectric constants. For these reasons, silicon dioxide was initially replaced by OSG and subsequently by porous OSG. These materials, which have a lower permittivity than SiO_2_, are referred to as low-*k* dielectrics [11].

The inverse relationship between Young’s modulus (YM) and the porosity of the material, which in turn affects mechanical strength, remains a significant challenge. High YM values, exceeding 5 GPa [12], are essential for the successful application of porous dielectrics in IC manufacturing [13]. Another significant issue associated with the porous structure is the adsorption of moisture, which subsequently deteriorates the dielectric properties of the material [14,15]. Therefore, to enhance hydrophobicity, it is essential to modify materials using various precursors that contain chemical groups capable of imparting high hydrophobicity. The most effective modification method involves using precursors with terminal methyl (–CH_3_) groups that exhibit thermal stability suitable for application in both back-end-of-line (BEOL) and front-end-of-line (FEOL) technology processes, specifically at temperatures ranging from 400 to 430 °C [16,17]. The modification of the pore surface by Si–CH_3_ leads to increased silicate hydrophobicity and a reduction in the dielectric constant (*k*), leakage currents, and breakdown field enhancement [13,18,19,20,21].

Copolymers derived from tetraethoxysilane (TEOS) and methyltriethoxysilane (MTEOS) have gained significant popularity in sol–gel synthesis. The incorporation of the MTEOS precursor into the copolymer composition enhances hydrophobicity and decreases the dielectric constant [22]. This effect is attributed to the presence of a terminal methyl group in the precursor molecule. Surface hydrophobicity depends on the number of introduced Si–CH_3_ bonds and residual Si–OH bonds present on the surface [23]. To achieve sufficient hydrophobicity, the content of MTEOS must be increased. However, an increase in the methyl-containing precursor leads to a reduction in the connectivity of the Si–O–Si network and, consequently, to a reduction in mechanical properties [24]. For this reason, it is challenging to strike a balance between hydrophobicity and film strength when using these two precursors. Consequently, there is a need to explore alternative methods for producing OSGs.

It is well established that dimethyl precursors, such as divinyldimethylsilane (DVDMS), hexamethylcyclotrisiloxane (HMCTSO), octamethylcyclotetrasiloxane (OMCTS), decamethylcyclopentasiloxane (DMCPSO), dimethyldimethoxysilane (DMDMOS), dimethyldioxosilycyclohexane (DMDOSH), divinylsiloxane-bis-benzocyclobutene (DVS-BCB), and dimethyl-dioxiranyl-silane (DMDORS), among others, are used in plasma-enhanced chemical vapor deposition (PECVD) [16,25]. Despite the significant potential and achievements associated with the incorporation of dimethyl groups, it is evident that certain precursors, such as HMCTSO, OMCTS, or DMCPSO, cannot achieve crosslinking without first degrading a portion of their dimethyl groups through thermal and/or ultraviolet (UV) radiation. This indicates a sacrificial role for methyl groups in PECVD, in contrast to chemical solution deposition (CSD). Furthermore, there are limited references in the literature regarding the application of precursors like DEDMS or DMDMOS in CSD technology. For instance, C.Y. Wang displays dimethyls together with monomethyls in the structure of a commercial film-forming solution of MSQ material from Tokyo Ohka Kogyo (TOK) Co. Ltd., Kanagawa, Japan [26]. However, no additional publicly available information, including patents, has been found regarding the composition of this solution, the study of films derived from it, or its widespread applications.

In this study, we aimed to compare MTEOS, which contains a single terminal group ≡Si–CH_3_, with diethoxydimethylsilane (DEDMS), which features two terminal methyl groups attached to a single silicon atom =Si(–CH_3_)_2_, representing a terminal dimethyl group. Several research groups have reported that the DEDMS precursor exhibits high hydrophobicity due to the presence of these two terminal methyl groups [27,28]. DEDMS-based copolymers are widely used in hybrid materials, including bulk materials, to improve hydrophobicity [29,30,31]. However, the properties of organosilica films produced from precursors containing terminal dimethyl groups have not been extensively investigated. The objective of this research is to provide a comprehensive and detailed analysis of organosilica films with varying concentrations of monomethyl and dimethyl groups. This analysis will involve a comparative examination of the chemical composition, microstructure, hydrophobic characteristics, mechanical properties, and electrical properties of the materials.

## 2. Materials and Methods

Film samples were prepared from film-forming solutions using CSD techniques. Two copolymers were selected for the experiments. The first copolymer was synthesized from tetraethoxysilane (TEOS, 99.999%, Sigma-Aldrich, St. Louis, MO, USA) and methyltriethoxysilane (MTEOS, 99%, Sigma-Aldrich, St. Louis, MO, USA) via sol–gel reactions. Hydrolysis and copolymerization of TEOS and MTEOS result in a silicon oxide network with terminal methyl groups, as the Si–CH_3_ bond is not hydrolyzed. The second copolymer was also synthesized based on TEOS, but instead of MTEOS, a precursor with two terminal methyl groups, namely diethoxydimethylsilane (DEDMS, 97%, Acros Organics, Geel, Belgium), was used. Accordingly, this copolymer contains dimethyl rather than monomethyl groups. The precursor molecules are illustrated in Figure 1. The compositions of the methyl and dimethyl copolymers were chosen to maintain the same number of methyl groups per silicon atom (CH_3_/Si) in order to compare the effects of the spatial distribution of methyl groups. The CH_3_/Si ratio was 0.2, 0.6, and 1.0 for both solutions containing MTEOS (series ‘m’) and those with DEDMS (series ‘d’). Thus, the solution with a CH_3_/Si ratio of 1.0 was prepared using either MTEOS or a 1:1 molar ratio of TEOS to DEDMS (see Table 1).

Film-forming solutions were prepared by adding alkoxide precursors with silicon concentrations of 6.5 and 1.2 wt% for the relatively thick and thin films, respectively, to an acidified water–isopropanol mixture. The molar ratios were [TEOS + MTEOS]:H_2_O:HCl = [TEOS + DEDMS]:H_2_O:HCl = 1:4:0.002. The resulting mixture was heated at 60 °C for 3 h. Afterwards, the sacrificial surfactant Brij^®^ L4 (molecular weight 362, Sigma-Aldrich, St. Louis, MO, USA) was added at a concentration of 19 wt% to induce porosity in these copolymer films via evaporation-induced self-assembly (EISA) [32]. A more detailed description of the film-forming solution preparation procedure is provided in the Appendix A.

Relatively thin and thick films were spin-coated onto the polished side of p^+^-Si:B monocrystalline wafers with resistivities of approximately 0.005 and 12 Ω∙cm, respectively. Deposition was performed using a WS-650-8NPP spin coater (Laurell Technologies, Lansdale, PA, USA) at a rotation speed of 2500 rpm for 30 s, with an acceleration rate of 1000 rpm/s. Subsequently, they were soft-baked on a hot plate at 150 °C for 30 min, followed by a hard bake—heat treatment in a furnace at 400 °C for 30 min in air without purging. The heating and cooling rates were maintained at ~10 °C/min. Thin films on wafers with a resistivity of 0.005 Ω·cm were primarily used to estimate the *k* values, enhancing the accuracy of the capacitive method by generating a stronger signal relative to background parasitic noise and the inherent capacitances of the measurement system.

The thickness (*d*) and refractive index (RI, *n*) at λ = 632.8 nm of the films were measured using spectral ellipsometry (SE) with an SE 850 ellipsometer (Sentech, Berlin, Germany) at a beam incidence angle of 70° over a wavelength range of 300–800 nm. The calculations were performed within the framework of the three-layer Cauchy model: air/film/silicon. Based on the measured thickness values obtained after soft baking and hard baking, the film shrinkage (Δ*d*) during heat treatment was calculated using the following formula:(1)$$∆d=100\frac{d_{1}-d_{2}}{d_{1}},$$
where *d*_1_ is the thickness of the soft-baked film, and *d*_2_ is the thickness after the heat treatment at 400 °C.

The infrared spectra were recorded using a Nicolet 6700 Fourier-transform infrared (FTIR) spectrometer (Thermo Electron Corporation, Waltham, MA, USA) in transmission mode, with a resolution of 4 cm^−1^ and a minimum of 64 scans. To minimize noise in the spectra caused by the atmosphere, the spectrometer was intensively purged with dry, pure nitrogen. This purging process was conducted for at least 50 min, with an increased flow rate during the first half hour, ~50 standard cubic feet per hour (scfh), compared to the normal rate of 30 scfh.

The refractive index is used to calculate the relative or full porosity within the framework of the Lorentz–Lorenz model. In this model, the relative volume fraction of pores in the film is determined using the following expression [33,34]:(2)$$V_{full}=100\%\cdot \left(1-\frac{\left(n_{eff}^{2}-1\right)\cdot \left(n_{mat}^{2}+2\right)}{\left(n_{mat}^{2}-1\right)\cdot \left(n_{eff}^{2}+2\right)}\right),$$
where *n_eff_* is the RI of the porous film; *n_mat_* is the RI of the dense silicate film, equal to 1.465.

The open porosity, which is accessible to adsorbate molecules, and the pore radius distribution were evaluated using ellipsometric porosimetry (EP). In the porosimetry setup, the partial pressure of the adsorptive is precisely controlled by two mass flow controllers (MFCs) to regulate the composition of the vapor–gas mixture delivered to the sample through a specialized thick-walled nozzle designed to prevent exposure to the atmosphere. One component of the vapor–gas mixture is dry nitrogen. The second component is also dry nitrogen, which functions as a carrier gas that passes through a thermo-stabilized bubbler containing liquid isopropyl alcohol (IPA), which is utilized as the adsorbate. The volume fraction of pores filled with condensed adsorbate, adsorbed within the film’s open porous structure, was calculated using the modified Lorentz–Lorenz equation [33,34]:(3)$$V=100\%\cdot \left(\frac{n_{eff}^{2}-1}{n_{eff}^{2}+2}-\frac{n_{p}^{2}-1}{n_{p}^{2}+2}\right)/\left(\frac{n_{ads}^{2}-1}{n_{ads}^{2}+2}\right),$$
where *n_eff_* is the RI of the porous film, partially or completely filled with adsorbate molecules, *n_p_* is the RI of the film before adsorption (empty pores), and *n_ads_* is the RI of the liquid adsorbate (1.377 for IPA). The pore size was determined by analyzing the gradual filling (adsorption) and emptying (desorption) curves, using the Kelvin and Dubinin–Radushkevich formulas for meso- and micropore size distribution, respectively. The radius obtained from the adsorption isotherms corresponds to the size of the internal cavities, while the radius derived from the desorption isotherms reflects the interconnections between these cavities and the external environment.

For precise control via the digital protocol of MFCs, a custom-developed script for the SE software SpectraRay II (Sentech, Berlin, Germany) is used to automate the execution of the required measurements and to initiate the calculation of the resulting ellipsometric parameters. This script manages the operation of the spectral ellipsometer SE 850. Based on the accumulated measurement data, adsorption and/or desorption isotherms are subsequently constructed. The calculation of these isotherms is then performed using the aforementioned formulas to evaluate the parameters of the porous structure. These procedures are described in more detail in refs. [33,34].

Young’s modulus is a fundamental mechanical property used to evaluate the elastic strength of a film. The YM values are determined based on deformations induced by adsorption or desorption, exhibiting a more pronounced dependence. The YM values (*E*) are calculated by approximating several points from the experimental curve that describes the change in film thickness (*d*) as a function of the partial pressure (*P*/*P*_0_) of adsorbate vapors, using [35]:(4)$$d=d_{0}+\kappa \cdot ln(P/P_{0}),$$
where *d*_0_ represents the initial film thickness prior to the desorption process, specifically at *P*/*P*_0_ = 1. By fitting the experimental data points of the deformation isotherm to a logarithmic function (Equation (4)), we derive the coefficient *κ*, which is then used to calculate *E* [35]:(5)$$E=\frac{d_{0}RT}{\kappa V_{L}},$$
where *R* is the gas constant, *T* is the temperature in Kelvin, and *V_L_* is the molar volume of the adsorbate. This method enables the estimation of *E* values for micro- and mesoporous films, reaching up to 8 GPa.

In addition, to investigate volumetric hydrophobicity, the pores were gradually filled with water in 20 s/step (excluding measurement time) as the vapor–gas partial pressure (*P*/*P*_0_) was incrementally increased using the porosimetric setup (water was used as the adsorbate instead of IPA). The water-filled pore volume was calculated using the following formula:(6)$$V_{H2O\;filling}\approx \frac{V_{H2O}}{V_{full}},$$
where *V_H_*_2*O*_ is the volume fraction of pores filled with condensed water (see Equation (3)) at *P*/*P*_0_ ≈ 0.45. This value was selected because general standards for cleanrooms (ISO 14644-1) [36] regulate the relative humidity (RH) in IC production to be between 40% and 60%, typically 45% ± 5%.

The mechanical properties of the films were also evaluated using the PeakForce Quantitative Nanomechanics (PFQNM) semi-contact mode with a Dimension Icon atomic force microscope (Bruker, Billerica, MA, USA). AFM studies in the PFQNM mode were conducted under ambient conditions using RTESPA525 silicon probes, which have a stiffness constant of *k* = 200 N/m and a probe tip radius of *R* = 8 nm, in accordance with refs. [37,38].

To investigate surface hydrophobicity, a method for measuring the contact angle with water (WCA) was employed. For this purpose, a DSA25B WCA device (KRÜSS, Hamburg, Germany) was utilized. The WCA values were calculated as the average of at least three different points, with ten photographs taken at each point. All measurements were conducted at room temperature (~23 °C). Data processing was conducted using KRÜSS ADVANCE version 1.13.0.21301.

The permittivity (*k*) was determined from the measured capacitance data using the formula for a parallel-plate capacitor. This measurement was performed using a mercury probe (MDC 802B-150) with a contact diameter of ~790 µm and an MDC CSM/Win Semiconductor measurement system (Materials Development Corporation, Andover, MA, USA) that contains a 4284A LCR meter (Agilent, Santa Clara, CA, USA). The metal–insulator–semiconductor structure was formed by positioning a mercury point electrode on a film deposited on highly doped silicon samples. The permittivity was calculated based on the frequency dependence of the capacitance measured at zero bias voltage, within a frequency range from 1 kHz to 1 MHz.

## 3. Results and Discussions

Table 1 presents the names of the samples, their compositions, and the results obtained from SE measurements: thickness (*d*), RI (*n*) after a soft bake on a hot plate at 150 °C for 30 min, and after annealing in a furnace at 400 °C for 30 min. Additionally, it includes calculated film shrinkage (Δ*d*) and full porosity (*V_full_*). The crosslinking coefficient is defined as the number of Si–O–Si bonds present at specific precursor ratios, expressed by the formula:(7)$$x=\left(m_{Y}\cdot Y+m_{z}\cdot Z\right)/2,$$
where *m_Y_* and *m_Z_* represent the mole fractions of the methyl-containing precursors (MTEOS or DEDMS) and TEOS, respectively; *Y* and Z denote the number of oxygen atoms in the corresponding precursors. A higher value of *x* indicates a more extensively cross-linked system, which is expected to enhance the elastic stiffness of the film material. For example, silicon dioxide has the chemical formula SiO_2_, which indicates that the value of *x* is equal to 2.

**Table 1 polymers-17-02433-t001:** Calculated compositional parameters and results obtained from spectral ellipsometry of the monomethyl (series ‘m’) and dimethyl (series ‘d’) porous organosilica films prepared from tetraethoxysilane (TEOS) mixed with either methyltriethoxysilane (MTEOS) or diethoxydimethylsilane (DEDMS).

Sample Name	TEOS/ MTEOS Ratio	TEOS/ DEDMS Ratio	TEOS Mole Fraction	CH_3_/ Si Ratio	Cross- Linking Coefficient *x* in SiO*_x_*	Soft Bake 150 °C, 30 min		Hard Bake 400 °C, 30 min			
						*d* (nm) [±2]	*n* [±0.003]	*d* (nm) [±2]	*n* [±0.003]	Δ*d* (%) [±3]	*V_full_* (%) [±1]
02m	80/20	-	0.8	0.2	1.9	116	1.398	101	1.239	13	45
02d	-	90/10	0.9			149	1.308	129	1.253	13	42
06m	40/60	-	0.4	0.6	1.7	105	1.374	97	1.243	8	44
06d	-	70/30	0.7			105	1.417	94	1.275	10	37
10m	0/100	-	0	1.0	1.5	96	1.413	91	1.285	5	35
10d	-	50/50	0.5			102	1.444	93	1.304	9	31

In this table: *d*—thickness, *n*—refractive index, Δ*d*—shrinkage, *V_full_*—full porosity. The initial digits in the sample names (02, 06, 10) represent the number of methyl groups per silicon atom: 0.2, 0.6, and 1.0, respectively. The second and third columns display the ratios of TEOS to the methyl-containing precursors, which determine the composition of the films. The fifth column illustrates the ratios of methyl groups to silicon atoms. The crosslinking coefficient is determined by the number of Si–O–Si bonds present at specific precursor ratios, as outlined in Equation (7). Higher values of this coefficient indicate a more crosslinked system.

As illustrated in Table 1, both series of samples demonstrate a decrease in the RI as the annealing temperature increases from 150 °C to 400 °C. This phenomenon can be attributed to the thermal destruction of the porogen and the subsequent formation of a porous structure within the films. In addition, the samples from the ‘m’ series exhibit higher relative porosity values compared to those from the ‘d’ series, which can be partly attributed to the significant thermal shrinkage observed in the latter. It should be noted that all compositions of both types of films exhibit relatively low shrinkage, not exceeding 13%.

FTIR spectra presented on Figure 2 show that, although calculations indicate the same number of methyl groups per silicon atom in both the ‘m’ and ‘d’ series, the dimethyl samples exhibit a lower intensity of the methyl group peak (1280–1265 cm^−1^). This discrepancy is attributed to the lower thermal stability of dimethyl groups [39,40]. In the case of CH_3_/Si = 0.2, sample ‘d’ exhibits a more pronounced peak at a position of ~3500 cm^−1^ and contains a higher amount of hydrogen-bonded surface silanols, along with adsorbed water, compared to sample ‘m’. The narrow peak at ~3750 cm^−1^ is attributed to the peak associated with isolated –OH groups that are not involved in water adsorption or intermolecular interactions with each other [41,42]. Their appearance also indicates a strong correlation with Si–OH at ~950 cm^−1^, while the methyl group transforms to Si–OH during thermal destruction [43,44]. It is also worth noting the broadening of the Si(–CH_3_)_1,2_ peak, which may indicate partial destruction of the Si(–CH_3_)_2_ group (~1265 cm^−1^) [45,46,47] during heat treatment at 400 °C. The asymmetry of the Si(–CH_3_)_1,2_ peak (refer to the inset in Figure 2) indicates that some Si atoms have only a single methyl group (~1275 cm^−1^) [45,46,47] after annealing.

The higher quantity of uncondensed residual silanol groups in the dimethyl samples can be attributed to the lower functionality of the DEDMS precursor, which contains two ethoxy (–OEt) functional groups, in contrast to MTEOS, which has three OEt groups available for polycondensation.

The peak associated with the presence of Si–CH_3_ and/or Si(–CH_3_)_2_ at ~1270 cm^−1^ is not clearly distinguishable [48]. It seems that there should be a similar relative amount of these groups, adjusted for the number of Si–O–Si bonds, but there are about 1.6 times fewer methyl groups in the ‘d’ series samples. The most plausible explanation for this discrepancy is the broadening of this peak in the ‘d’ series spectra, coupled with the strengthening of the cage component, which is also shifted to the region of higher wavenumbers. Such overlapping of bands often leads to a decrease in the reliability of directly determination the area under the curve (AUC) using a linear local baseline, typically resulting in an underestimation of values obtained for less pronounced peaks. This is indirectly supported by the lack of noticeable differences in the region of C–H group vibrations at 3000–2900 cm^−1^ (see Table 2).

It is evident from Table 2 that the ratio of the network/cage absorption intensities is significantly higher for the ‘d’ series samples. The network component is characterized by long Si–O–Si chains (SiO_2_, i.e., *x* = 2; see Figure 3a) and is located within the range of 1100–1050 cm^−1^. A networked suboxide consists of long Si–O–Si chains with terminal groups, specifically SiO*_x_*, where 1 < *x* < 2 (see Figure 3b). This structure is observed within the range of 1050 to 1000 cm^−1^. In contrast, the cage component is associated with a short chain of Si–O–Si tending to create cyclical structures (see Figure 3c) and is situated within the range of 1250–1100 cm^−1^. The network/cage ratio provides an indirect assessment of the mechanical strength of the structure: higher values indicate a stronger structure [49,50,51]. To improve the visualization of changes, spectral regions were normalized to the cage (see Figure 3d), as previously demonstrated in ref. [50]. The network/cage ratio values correlate well with the YM values obtained through porosimetry and PFQNM AFM, as will be demonstrated later in this paper. This increase can be attributed to the greater amount of the TEOS precursor used, which promotes crosslinking. However, this also results in a side effect: increased shrinkage. Note that in the case of CH_3_/Si = 0.2, this effect is less pronounced due to the higher concentration of residual silanol groups. These terminal silanol groups also contribute to an increase in the networked suboxide component, resulting in a slight rightward shift (red shift) in the position of the maximum of the Si–O–Si band, as well as broadening it. The primary cause of this broadening is the significant contribution from the Si–O–Si network component formed by the TEOS precursor. At the same time, the cage component displays a large width, indicating a disruption of homogeneity or long-range order within the material’s microstructure. In contrast, sample 10m represents the so-called methylsilsesquioxane (MSSQ), which tends to form homogeneous ladder-like or cage-like structures [52]. This results in narrower cage and networked suboxide components, which are clearly visible in the spectra. Additionally, the position of the common peak of the broad Si–O–Si band, which correlates with the Si–O–Si bond angle shown in Figure 3, slightly influences the position of the Si(–CH_3_)_1,2_ peak because the methyl groups are directly bonded to the silicon atom. Therefore, when the broad Si–O–Si band shifts, the Si(–CH_3_)_1,2_ peak, located on the shoulder of this band, shifts in the same direction (see Table 2).

The pore structure characteristics of the two copolymer films show a difference. As shown in Figure 4a, at a low precursor-to-TEOS ratio, the ‘m’ and ‘d’ series samples exhibit fairly similar pore radius distributions, with a slightly larger average pore radius for the ‘m’ samples, which may be attributed to the increased value of open porosity [54]. As the concentration of the TEOS precursor in the copolymers decreases, the differences in pore radius distributions between the ‘m’ and ‘d’ series samples become increasingly pronounced. Specifically, the ‘d’ series samples, which contain dimethyl groups, exhibit pores with a smaller radius. This phenomenon can be attributed to an increased crosslinking density, and it becomes more pronounced as the CH_3_/Si ratio increases. As is known from ref. [35], a more crosslinked system is indicated by a less steep thickness isotherm (see the bottom line of Figure 4). Consequently, the ‘d’ series samples exhibit a higher YM value compared to the ‘m’ series. In the monomethyl copolymer films, the hysteresis is less pronounced, indicating the quasi-cylindrical nature of the pores. It is also important to note that, in the case of CH_3_/Si = 0.6, the ‘d’ series samples continue to exhibit a bimodal distribution, while the ‘m’ series samples lose their microporous structure. This loss may be attributed to smaller pores merging into larger ones, resulting in an increase in the average pore radius. More detailed data obtained from porosimetry are presented in Table 3.

It is well established that an increase in porosity leads to a decrease in YM. The relationship between YM and porosity is typically described by the Phani–Niyogi equation [55]. Therefore, to compare the YM values of samples with different porosities, we normalized the obtained YM values to a specific porosity level, such as 30%. This value is established because ultra-low-*k* (ULK) films must have *k* < 2.5 [21], which corresponds to a porosity greater than 30% in SiO_2_. Additionally, NCS (nano-clustering silica) from TOK, one of the most successful spin-on ULK dielectrics used in mass IC production, exhibits approximately this level of porosity [56]. This normalization can be accomplished using the aforementioned equation [55]:(8)$$E_{eff}=E_{s}{\left(1-\phi /\phi_{0}\right)}^{n},$$
where *E_eff_* is the effective elastic modulus of the porous material, *E_s_* represents the elastic modulus of the solid phase (skeleton), *ϕ* denotes porosity expressed as a fraction, and *ϕ*_0_ is the critical porosity at which *E**_eff_* = 0. All three empirical parameters—*ϕ*_0_ ≈ 0.84 [from 0.8 to 0.99], *n* ≈ 2.9 [from 2 to 3], and *E_s_* ≈ 24.3 [from 10 to 40]—were determined by numerically fitting the model curve to the experimental data points, which are represented by green ‘×’ symbols in Figure 5b. These points correspond to Sample 06m, with porosities generated by varying the surfactant concentration from 20 wt% to 50 wt%. Figure 5b provides a detailed representation of the normalization procedure.

Thus, despite the results obtained from non-porous copolymer films with similar compositions [28], which indicate a decrease in hardness and YM when replacing MTEOS with DEDMS, our EP results present a contrasting perspective. Moreover, this dynamic is evident in the range of CH_3_/Si > 0.2. One possible reason for this discrepancy is that ref. [28] used materials in which MTEOS was only partially replaced by DEDMS. Additionally, they employed different evaluation methods, such as nanoindentation tests, without normalizing for porosity. However, it is well known that dense OSG films contain ultramicropores that are difficult to detect; nevertheless, these pores can account for up to ~15% of the full porosity [57].

Table 4 shows that the PFQNM AFM measurements confirm an increase in YM for samples with terminal dimethyl groups. However, this increase is counterbalanced by significant measurement errors, especially in samples with low YM values (CH_3_/Si = 1.0). Additionally, the values obtained by PFQNM AFM are slightly higher than those obtained by EP. A detailed explanation of this difference is provided in our previous work [37].

Nanoindentation was not performed due to significant errors observed in measurements on relatively thin films. The substrate effect also has a substantial influence; to minimize this, the recommended penetration depth for soft, porous films on very hard substrates, such as silicon, should not exceed 10% of the film thickness [58]—approximately 50 nm for films ~0.5 μm thick. Achieving such a shallow penetration requires a load of less than 0.1 mN, which is too low to ensure reliable results. Conducting these measurements necessitates the use of ultrananoindenters. However, to the authors’ knowledge, such instruments remain very rare.

To illustrate the differences in YM values and network/cage ratios, corresponding simulations were conducted and are presented in Table 5. This speculative 2D model schematically illustrates the global behavior of the system, based on the strong idealization that methyl groups are concentrated in the upper-left corner of the network fragment, with minimal residual uncondensed silanol groups.

Examining Table 5, we observe that as the MTEOS content increases relative to the TEOS precursor, the ratio of fully crosslinked silicon atoms to the total number of silicon atoms (ψ) decreases. This trend ultimately leads to the complete disappearance of these atoms when only MTEOS is used. In the ‘d’ series samples, the presence of two methyl groups attached to silicon also results in a decrease in the number of fully crosslinked Si atoms as the concentration of the DEDMS precursor increases. However, the DEDMS precursor retains more of these atoms than the ‘m’ series does. Thus, the increased number of these atoms correlates positively with the higher YM values observed in the ‘d’ series samples. In reality, the crosslinking coefficient *x* does not accurately represent the degree of crosslinking, which is determined solely by the precursor responsible for generating fully crosslinked silicon atoms within the film; that is, the mole fraction of TEOS ≡ ψ.

Water contact angle measurements are shown in the inset of Figure 6. The ’d’ samples, which had the dimethyl group, showed lower levels of hydrophobicity. This is likely due to the decreased number of methyl groups in the films that underwent heat treatment. It is also important to note that at a concentration of 100 mol% MTEOS, the film becomes hydrophobic, as its contact angle exceeds 90° [59]. In contrast, a film composed of 100 mol% DEDMS has a contact angle that is at the threshold of hydrophobicity. The measured WCA values are as follows: 02m—34.8° ± 0.4°, 06m—75.4° ± 0.6°, 10m—112.2° ± 1.7°, 02d—38.5° ± 3.0°, 06d—62.2° ± 1.3°, and 10d—89.5° ± 1.0°. However, for porous materials, bulk hydrophobicity is of greater interest than surface hydrophobicity. Notably, bulk hydrophobicity (see Figure 6) correlates strongly with WCA measurements, indicating that surface effects are minimal. Interestingly, both methods show similar or slightly higher hydrophobicity for the dimethyl film compared to the monomethyl film at CH_3_/Si = 0.2, whereas FTIR reveals the most significant differences favoring the monomethyl sample (see Figure 2).

The dielectric constant increases in porous organosilica films when water is absorbed because water has a very high permittivity (*k* ≈ 81). Figure 7 shows the dependence of the dielectric constant on the CH_3_/Si content in copolymer films. An increase in the amount of methyl content leads to a decrease in the dielectric constant for both series. A higher number of methyl groups and lower silanol content in the ‘m’ series lead to lower values of the dielectric constant. Figure 7b shows the relationship between permittivity and full porosity as described by the Clausius–Mossotti equation [48,60]:(9)$$\frac{k_{p}-1}{k_{p}+2}=\left(1-V\right)\cdot \frac{k_{s}-1}{k_{s}+2}$$
where *k_p_* is the effective dielectric constant of the composite material, *k_s_* is the dielectric constant of the film skeleton, and *V* is the relative pore volume *V_pore_*/*V_film_*, representing the full porosity *V_full_*.

Samples 02d, 02m, and 06d lie above the conventional boundary for SiO_2_ (yellow dashed curve), indicating that they fall outside the green-shaded region due to their high concentration of surface silanols. The effect of water adsorption is most pronounced in films prepared from pure TEOS, which exhibit higher leakage currents and capacitance that strongly depend on frequency. For this reason, the dielectric constant can reach unusually high values (*k* > 8) [61], compared to ~2.3 for 40% porous SiO_2_ (see Figure 7b). To compare the *k* values of samples with varying porosities, we normalized the obtained *k* values to a specific porosity level, such as 30%, as shown in Figure 7.

Figure 8 summarizes the key parameter values: pore radius, YM, and dielectric constant.

As shown, the ‘06d’ sample demonstrates the best performance across nearly all parameters. Therefore, the distribution and number of methyl groups within porous organosilica films can significantly impact the structure, including pore size, as well as surface chemistry, elastic properties, and dielectric constant. This provides a potential avenue for the development of films with tailored properties for diverse applications.

## 4. Conclusions

Organosilica films are versatile materials with a wide range of applications, including electronics, optics, catalysis, drug delivery, and many others. In electronics, porous organosilica films are valued for their low dielectric constant. Most applications require high hydrophobicity because water adsorption degrades film properties. The most common way to provide hydrophobicity is to protect the film surface with methyl groups. However, this approach reduces the connectivity of the silicon–oxygen network and, as a result, degrades the mechanical properties of the films. To find an optimal balance between these properties, we compare porous organosilica films containing monomethyl and dimethyl groups attached to the silicon atom.

In sol–gel synthesis, a hybrid organic–inorganic material is produced using two components: one is silicon alkoxide TEOS, which forms only a Si–O–Si network, while the other is a methyl-modified alkoxide, MTEOS, which contains Si–CH_3_ bonds that do not undergo hydrolysis and remain intact in the final material. This set of experiments is referred to as the ‘m’ series. To prepare films containing Si(–CH_3_)_2_ bonds, we used diethoxydimethylsilane (DEDMS) instead of MTEOS, which is used in the ‘m’ series. The film composition was varied by adjusting the ratio of the two components, quantified as the number of methyl groups per silicon atom: 0.2, 0.6, and 1.0. The non-ionic surfactant Brij^®^ L4 was used to provide porosity by self-assembly. The study included characterization by FTIR, spectral ellipsometry, ellipsometric porosimetry (EP), PFQNM AFM, water contact angle (WCA) measurements, and dielectric constant evaluations.

The study revealed that, although both the ‘m’ and ‘d’ series samples have the same ratio of methyl groups to silicon atoms, the ‘m’ series films contain a greater total number of methyl groups. This finding is supported by the FTIR spectra. Dimethyl groups are susceptible to thermal destruction, which subsequently impacts the processes of secondary condensation and crosslinking. Additional crosslinking influences the porous structure by strengthening the pore walls, as evidenced by the increased Young’s modulus values observed in the dimethyl films.

The porosimetry study indicates that the ‘d’ series samples exhibit a smaller average pore size compared to the ‘m’ series samples. This observation is consistent with the observed decrease in open porosity. The reduced average pore size can also be attributed to increased crosslinking that occurs during the thermal destruction of dimethyl groups.

An important observation is the higher concentration of silanol groups in the ‘d’ series samples, which influences water adsorption and subsequently leads to increased dielectric constant values. Additionally, the higher dielectric constant can be partly attributed to reduced porosity. The combination of factors—including an increased number of Si–OH groups, a reduced number of methyl groups, and lower open and total porosity values compared to the ‘m’ series samples—results in decreased hydrophobicity of the samples derived from the precursor containing the terminal dimethyl group, as confirmed by water EP and WCA measurements.

A speculative model of the crosslinking behavior in monomethyl and dimethyl copolymer films indicates that, at equivalent ratios of methyl groups to silicon atoms, dimethyl films exhibit a higher degree of crosslinking. This observation correlates with the Young’s modulus values obtained through both ellipsometric porosimetry and AFM methods. It is also indirectly supported by quantitative FTIR analysis, specifically the broad Si–O–Si absorption band.

Thus, copolymer films containing dimethyl groups provide a higher degree of crosslinking and, therefore, higher mechanical properties compared to films with an equivalent amount of methyl groups. However, the lower thermal stability of the dimethyl bonds results in reduced hydrophobicity and related properties, such as the dielectric constant. These data will increase our knowledge of organic–inorganic hybrid films and their structural and physical properties. The choice between these two components for practical applications depends on the specific requirements of the particular application.

## Figures and Tables

**Figure 1 polymers-17-02433-f001:**
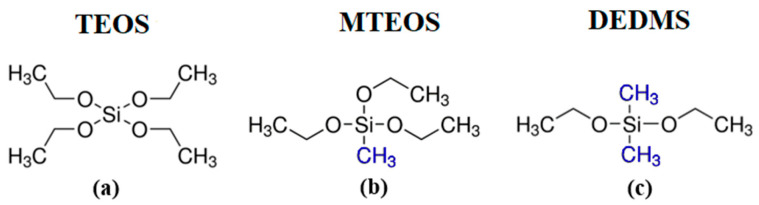
Structures of precursor molecules: (**a**) tetraethoxysilane (TEOS), (**b**) methyltriethoxysilane (MTEOS), and (**c**) diethoxydimethylsilane (DEDMS).

**Figure 2 polymers-17-02433-f002:**
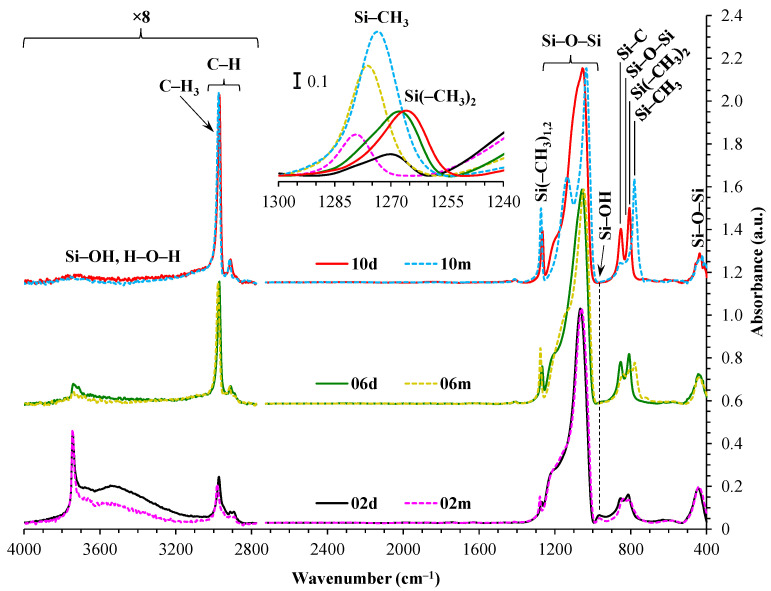
FTIR spectra of the monomethyl (samples ‘m’) and dimethyl (samples ‘d’) porous organosilica films, normalized to the amplitude of the broad Si–O–Si band. The inset displays the Si–CH_3_ peak, normalized for thickness and the fraction of open porosity, and multiplied by a factor of 1000. The initial digits in the sample names (02, 06, 10) represent the number of methyl groups per silicon atom: 0.2, 0.6, and 1.0, respectively.

**Figure 3 polymers-17-02433-f003:**
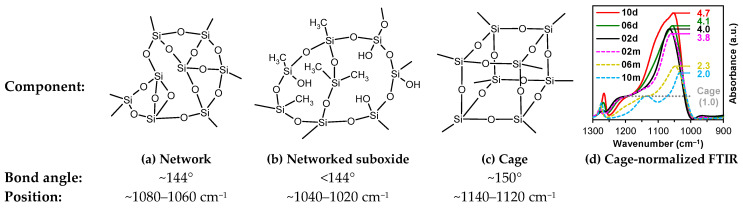
Components of a broad Si–O–Si band, which include the network (**a**), networked suboxide (**b**), and cage (**c**) structures, observed in the range of 1300 to 1000 cm^−1^ (**d**) in the cage-normalized FTIR spectra of the monomethyl (samples ‘m’) and dimethyl (samples ‘d’) porous organosilica films. The initial digits in the sample names (02, 06, 10) represent the number of methyl groups per silicon atom: 0.2, 0.6, and 1.0, respectively. Sections (**a**–**c**) have been partially redrawn from ref. [53]. The bond angle and position data were taken from ref. [49].

**Figure 4 polymers-17-02433-f004:**
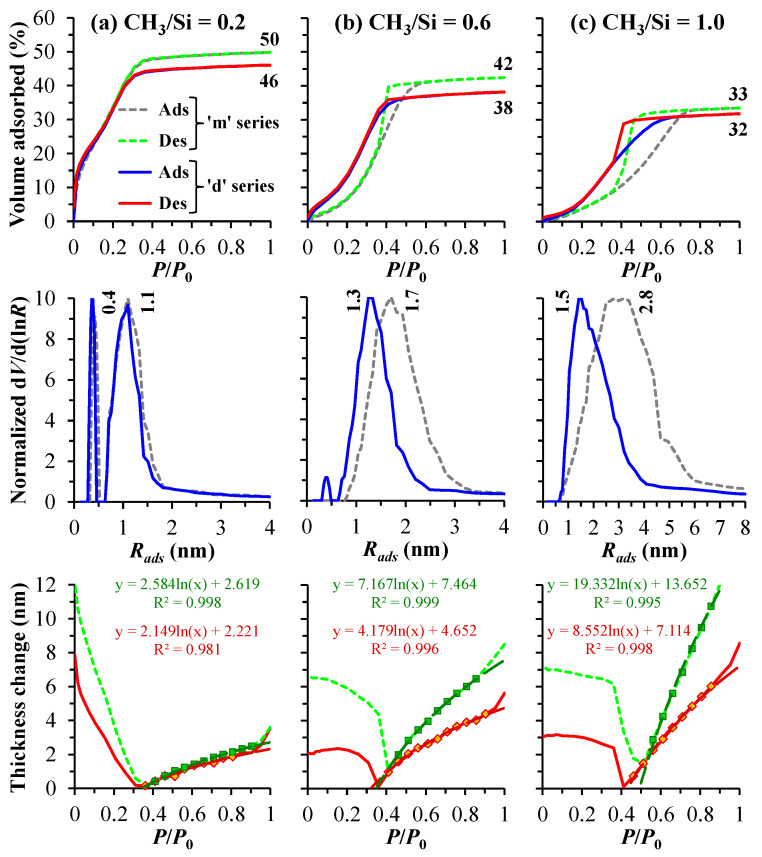
Adsorption (Ads) and desorption (Des) isotherms as functions of the partial pressure (*P*/*P*_0_) of the vapor–gas mixture, which indicate the degree of pore filling and emptying by the adsorbate (isopropyl alcohol, top line), the pore radius (*R_ads_*) distribution derived from the Ads branch (middle line), and the change in thickness during desorption (bottom line) for samples with varying numbers of methyl groups per silicon atom (CH_3_/Si) of 0.2 (**a**), 0.6 (**b**), and 1.0 (**c**). The series ‘m’ and ‘d’ represent monomethyl and dimethyl porous organosilica films, respectively. In the bottom line, markers represent experimental data points of the deformation isotherm (Des branch), which is modeled using a logarithmic function (Equation (4)), as indicated by the long-dashed lines.

**Figure 5 polymers-17-02433-f005:**
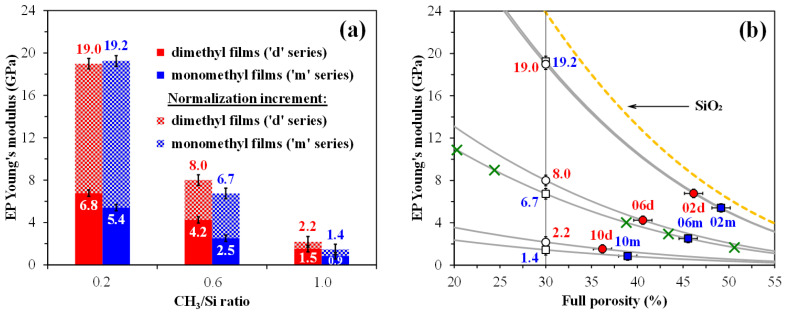
Relationship between Young’s modulus (YM) and the number of methyl groups per silicon atom (CH_3_/Si) in panel (**a**), as well as the correlation between YM and full porosity in panel (**b**). The gray curves in panel (**b**) are plotted according to the Phani–Niyogi equation [55], with *ϕ*_0_ ≈ 0.84 and *n* ≈ 2.9, intersecting the corresponding measured data points of the samples under investigation. The yellow dashed curve represents SiO_2_ with *E_s_* ≈ 86 GPa, using the same *ϕ*_0_ and *n* coefficients. Experimental data points, denoted by the green symbol ‘×’, correspond to Sample 06m, with porosities generated by varying the surfactant concentration (20–50 wt%). The normalization increment in panel (**a**) refers to adjusting the obtained YM values to a specific porosity level, such as 30%. The initial digits in the sample names (02, 06, 10) represent the number of methyl groups per silicon atom: 0.2, 0.6, and 1.0, respectively.

**Figure 6 polymers-17-02433-f006:**
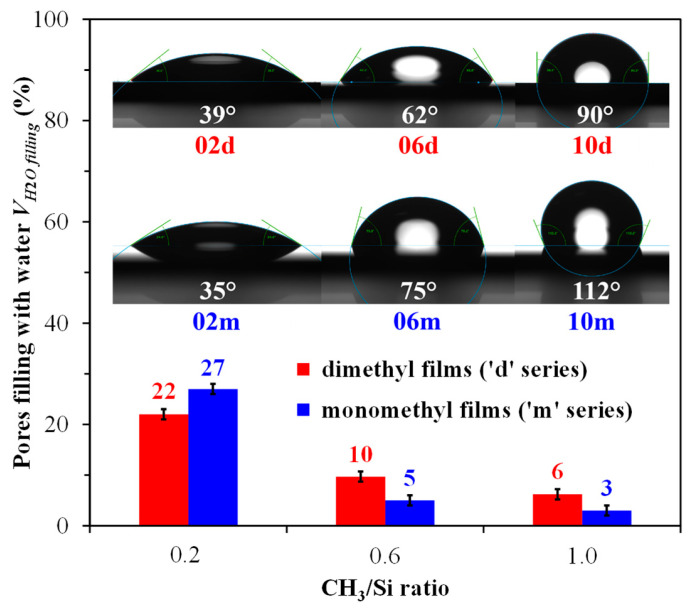
Hydrophobicity of the monomethyl (‘m’ series) and dimethyl (‘d’ series) porous organosilica films, presenting both volumetric data obtained from water ellipsometric porosimetry, indicated by pore filling with water at a partial pressure of *P*/*P*_0_ ≈ 0.45, and surface data measured by water contact angle (inset). The initial digits in the sample names (02, 06, 10) represent the number of methyl groups per silicon atom: 0.2, 0.6, and 1.0, respectively.

**Figure 7 polymers-17-02433-f007:**
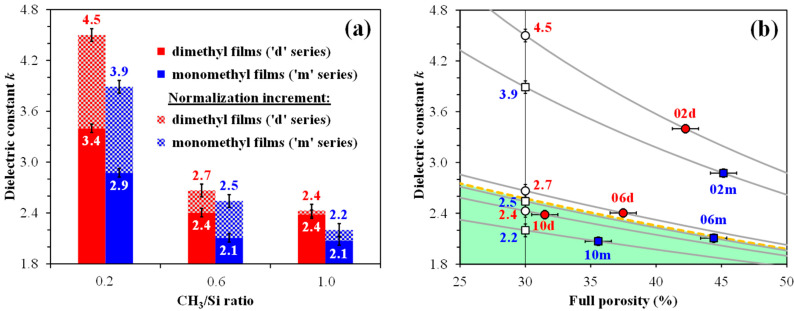
Relationship between the dielectric constant (*k*) and the number of methyl groups per silicon atom (CH_3_/Si) in panel (**a**), as well as the relationship between *k* and full porosity in panel (**b**). The gray curves in panel (**b**) are plotted using the Clausius–Mossotti equation [48], intersecting the corresponding measured data points of the samples under investigation. The yellow dashed curve represents SiO_2_ (*k_s_* ≈ 3.9). The normalization increment in panel (**a**) refers to adjusting the obtained *k* values to a specific porosity level, such as 30%. The initial digits in the sample names (02, 06, 10) represent the number of methyl groups per silicon atom: 0.2, 0.6, and 1.0, respectively.

**Figure 8 polymers-17-02433-f008:**
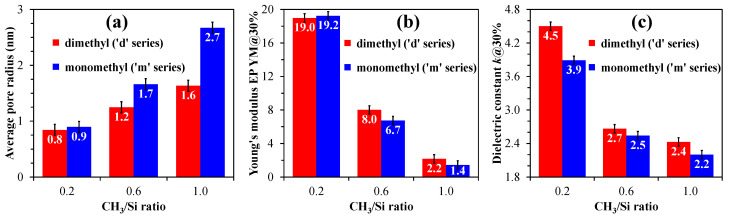
Values of key parameters: average pore radius (**a**); Young’s modulus (YM), derived from ellipsometric porosimetry (EP) data and normalized to a 30% porosity level (**b**); and the evaluated *k* value, also normalized to a 30% porosity level (**c**), for monomethyl (‘m’ series) and dimethyl (‘d’ series) porous organosilica films, which vary in the number of methyl groups per silicon atom (CH_3_/Si ratio).

**Table 2 polymers-17-02433-t002:** Quantitative analysis of FTIR spectra of the monomethyl (‘m’ series) and dimethyl (‘d’ series) porous organosilica films.

Sample Name	Position of the Absorption Peak/Band (cm^−1^)		Area Under the Absorption Peak/Band, Related to the Broad Band Si–O–Si (×100)					Network/Cage Ratio
	Si(–CH_3_)_1,2_	Si–O–Si	Si–OH, H–O–H	C–H	C–H_3_	Si(–CH_3_)_1,2_	Si–OH	
02m	1279	1060	5.14	0.93	0.33	0.70	0.44	3.8
02d	1269	1065	9.57	1.02	0.37	0.37	0.75	4.0
06m	1276	1048	0.64	2.02	1.23	2.91	0.02	2.3
06d	1268	1056	1.22	2.08	1.23	1.65	0.09	4.1
10m	1274	1034	0.54	3.53	2.34	5.75	-	2.0
10d	1266	1054	0.55	3.14	1.88	2.91	0.01	4.7

The initial digits in the sample names (02, 06, 10) represent the number of methyl groups per silicon atom: 0.2, 0.6, and 1.0, respectively.

**Table 3 polymers-17-02433-t003:** Key ellipsometric porosimetry (EP) parameters of the thicker porous organosilica films: monomethyl (‘m’ series) and dimethyl (‘d’ series).

Sample Name	*d* (nm) [±2]	*n* [±0.003]	*n_s_* [±0.005]	*V_open_* (%) [±2]	*V_full_* (%) [±1]	*R_ads_*	*R_des_*	⟨*R_ads_*⟩	Δ*_m_R_ads_*	EP YM	EP YM@30%
						(nm) [±0.1]				(GPa) [±0.3]	(GPa) [±0.5]
02m	465	1.221	1.472	50	49	0.4/1.1	0.4/1.1	0.9	0.6	5.4	19.2
02d	453	1.238	1.471	46	46	0.4/1.1	0.4/1.1	0.8	0.6	6.8	19.0
06m	558	1.238	1.436	42	46	1.7	1.7	1.7	1.1	2.5	6.7
06d	547	1.261	1.444	38	41	1.3	1.3	1.2	0.7	4.2	8.0
10m	507	1.269	1.421	33	39	2.8	1.9	2.7	2.8	0.9	1.4
10d	402	1.282	1.431	32	36	1.5	1.7	1.6	1.7	1.5	2.2

In this table: *d*—thickness, *n*—refractive index, *n_s_*—skeleton refractive index, *V_open_*—open porosity, *V_full_*—full porosity, *R_ads_*—pore radius during adsorption, *R_des_*—pore radius during desorption, ⟨*R_ads_*⟩—average pore radius during adsorption, Δ*_m_R_ads_*—half-width of the pore radius distribution during adsorption, EP YM—Young’s modulus evaluated by EP, EP YM@30%—Young’s modulus evaluated by EP and recalculated to account for 30% porosity using the Phani–Niyogi equation [55]. The initial digits in the sample name (02, 06, 10) represent the number of methyl groups per silicon atom: 0.2, 0.6, and 1.0, respectively.

**Table 4 polymers-17-02433-t004:** Young’s modulus (YM) data evaluated by AFM and recalculated to account for 30% porosity, as previously described, in the monomethyl (‘m’ series) and dimethyl (‘d’ series) porous organosilica films.

CH_3_/Si Ratio	AFM YM@30% (GPa) [>±0.5]	
	Series ‘m’	Series ‘d’
0.2	19.8	25.8
0.6	7.8	8.6
1.0	2.8	3.0

**Table 5 polymers-17-02433-t005:** Idealized speculative model of crosslinking for the monomethyl (‘m’ series) and dimethyl (‘d’ series) porous organosilica films, which vary in the number of methyl groups per silicon atom (CH_3_/Si ratio).

CH_3_/Si Ratio	Series ‘m’	Series ‘d’
0.2	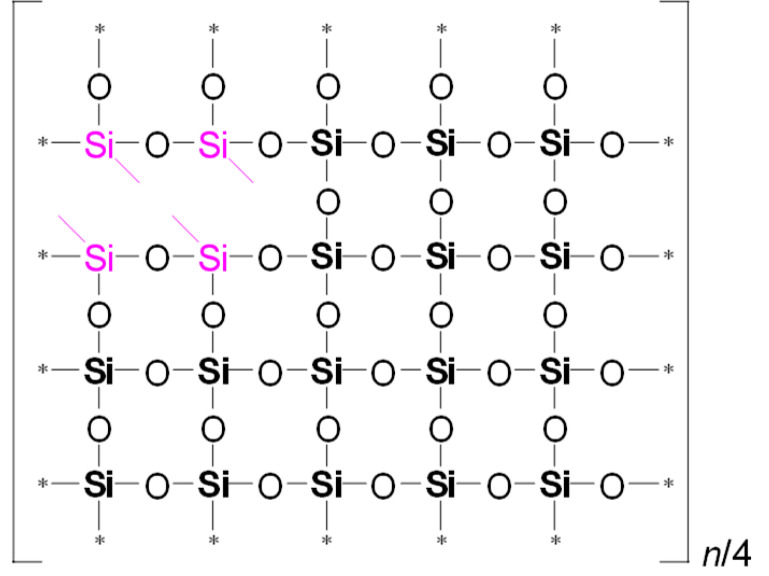 ψ = 16/20 = 0.8	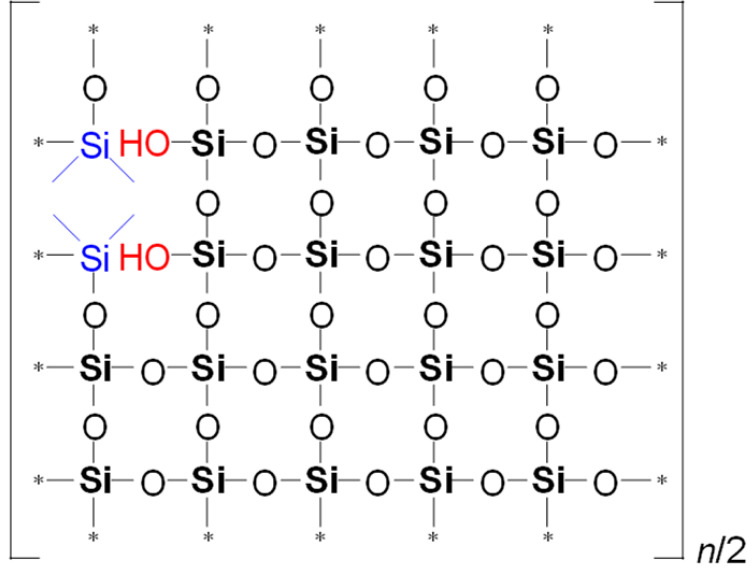 ψ = 18/20 = 0.9
0.6	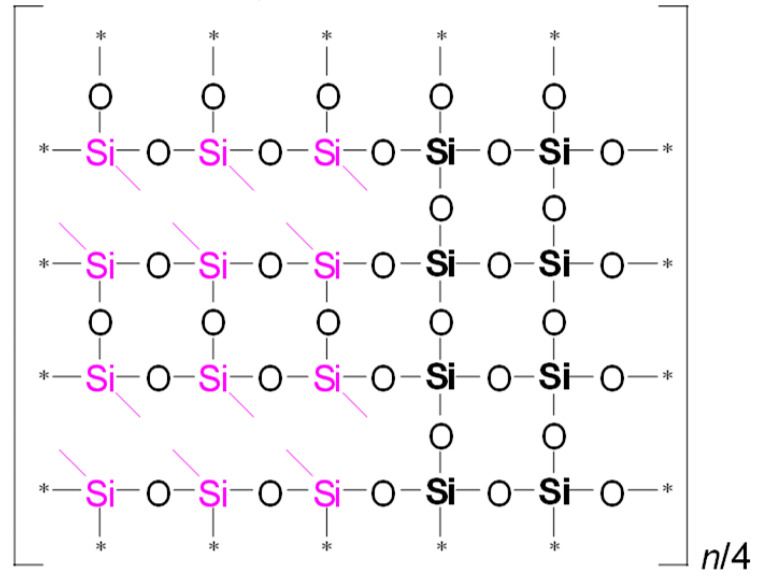 ψ = 8/20 = 0.4	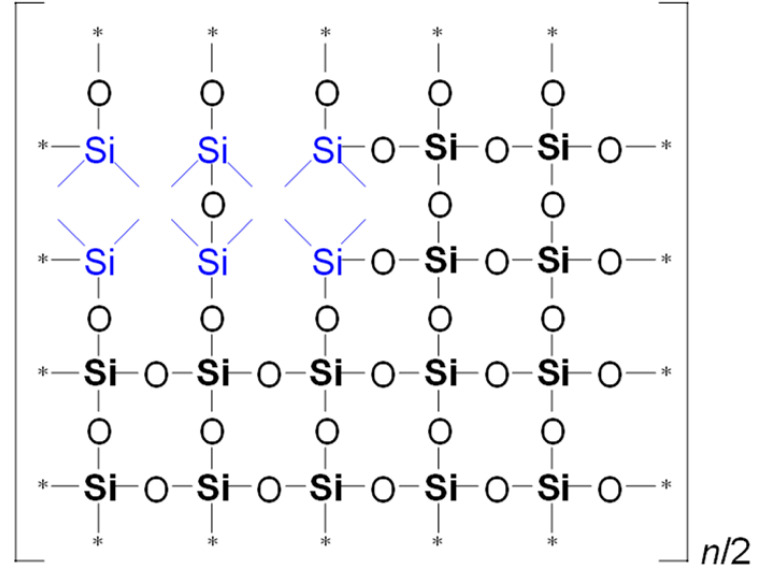 ψ = 14/20 = 0.7
1.0	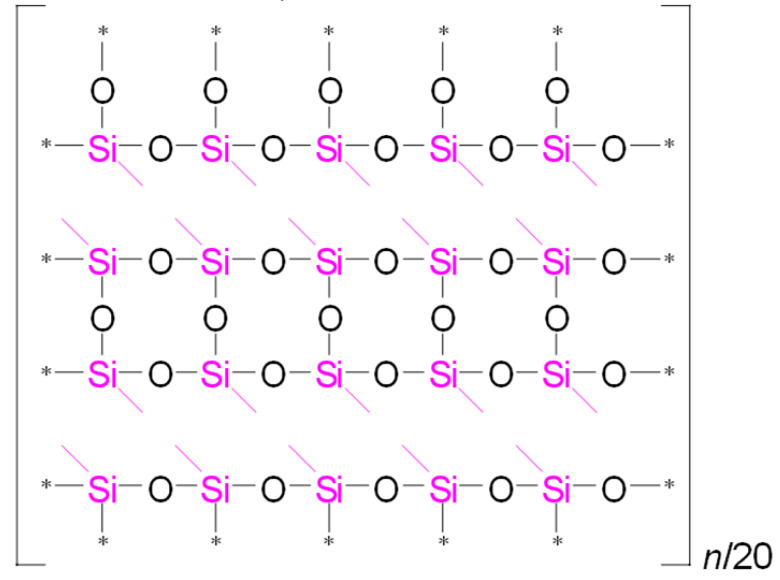 ψ = 0/20 = 0	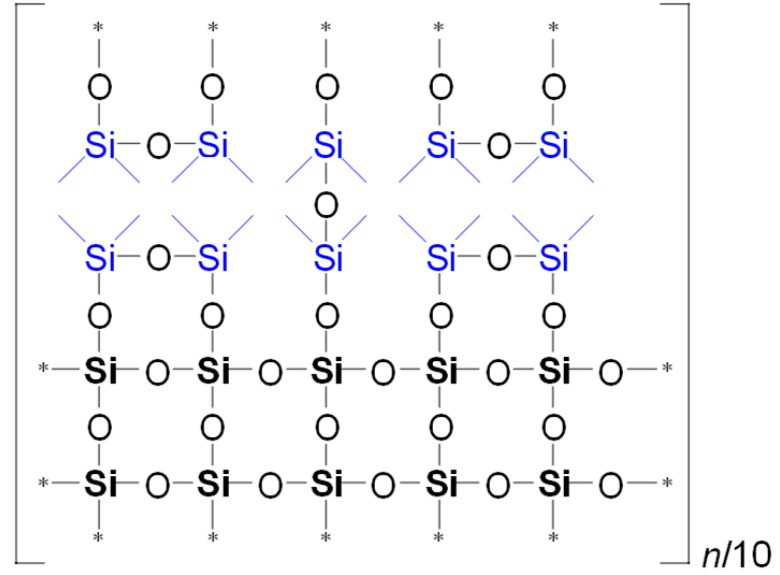 ψ = 10/20 = 0.5

In this table, ψ is the ratio of fully crosslinked Si atoms to the total number of Si atoms. The asterisk (*) denotes atoms at the boundary of the simulation cell, representing the continuation of the periodic structure. The ratio *n*/## represents the stoichiometry of the hybrid material, indicating the number of structural units required to model the composition. For example, for Sample 02m with an *n*/4 ratio, a fragment containing 20 Si atoms must include four minimal stoichiometric groups, each consisting of 5 base units (since 20/4 = 5): one unit derived from MTEOS and four units from TEOS. Consequently, the model fragment contains 4 units derived from MTEOS and 16 units derived from TEOS. Si–CH_3_ bonds of MTEOS are shown in pink, while Si(–CH_3_)_2_ bonds of DEDMS are depicted in blue.

## Data Availability

Data are contained within the article. The raw data presented in this study are available on request from the authors.

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
