# Peer review of "Porous Organosilica Films: Is It Possible to Enhance Hydrophobicity While Maintaining Elastic Stiffness?"

_polymers, 2025, doi:10.3390/polym17172433_

Round 1

Reviewer 1 Report

Comments and Suggestions for Authors

The article submitted for review contains three significant lacunae.

1. The authors do not present the exact procedure for synthesising the materials, which is necessary for the experiment to be replicated.

2. The lack of indication and testing of material properties using environmental samples could enhance the importance of the research.

3. None of the designed materials are composed of only one component for comparison (TEOS, MTEOS or DEDMS). Preparing these polymers would provide valuable insight into how the composition of a material influences its physicochemical properties.

The manuscript is good-prepared, with a clear introduction and correct visualisation of the results. However, there is no description of the synthetic procedure for obtaining the materials. As the entire work is based on the analysis of synthesised materials, the absence of a description of the synthesis should result in the rejection of the article.

Author Response

Please see the attached list of revisions and the authors' responses to Reviewer 1's comments.

Reviewer 2 Report

Comments and Suggestions for Authors

Porous organosilica films have found important applications in various fields, particularly in microelectronics, where high-performance dielectric materials are required. However, tuning the balance between mechanical strength and bulk hydrophobicity – both critical for long-term performance – has been a major challenge. In this work, the authors systematically study how the ratio of terminal Si-Me groups influences these properties by incorporating monomethyl or dimethyl precursors during thin film preparation. They report that the dimethyl precursor could enhance the Young’s modulus but simultaneously reduces the hydrophobicity and dielectric performance. These findings expand our understanding of the structure-property relationships in organosilica films and provide valuable insights for the development of high-performance organosilica materials. Personally, I have some questions and concerns listed below:

  1. On page 3, lines 131-132, the film was heated at 400 °C under air. Is this a standard practice for such systems, or would an inert atmosphere be more appropriate to avoid potential oxidation?
  2. On page 4, additional details of the “custom-developed program” used for EP analysis should be provided to ensure reproducibility.
  3. According to Figure 2, within the 1000-1240 cm-1 region, sample 10d shows a broader peak comparing to sample 10m. But other samples do not exhibit such broadening. This trend should be explained. Additionally, in the inset, peaks assigned to either Si-CH3 or Si-(CH3)2 shift between samples without a clear correlation to precursor composition. The authors should discuss possible reasons.
  4. To better understand the surface chemical composition, I suggest performing XPS analysis as well. SEM and EDS are also highly recommended.
  5. On page 11, the authors stated that their experimental results contrast with Ref. 28. For a more clear and fair comparison, please clarify whether Ref. 28 used the same materials, calculation methods, and assumptions/normalizations.
  6. Detailed descriptions of the simulation methods used for Table 5 should be included for reproducibility.

I would recommend acceptance for publication as long as the aforementioned questions and concerns have been addressed appropriately.

Author Response

Please see the attached list of revisions and the authors' responses to Reviewer 2's comments.

Round 2

Reviewer 1 Report

Comments and Suggestions for Authors The manuscript is good-prepared, with a clear introduction and correct visualisation of the results. However, there is no detailed description of the synthetic procedure for obtaining the materials. As the entire work is based on the analysis of synthesised materials, the absence of a description of the synthesis should result in the rejection of the article.

Author Response

Comments 1: The manuscript is good-prepared, with a clear introduction and correct visualisation of the results. However, there is no detailed description of the synthetic procedure for obtaining the materials. As the entire work is based on the analysis of synthesised materials, the absence of a description of the synthesis should result in the rejection of the article.
Response 1: We thank the reviewer for their constructive comments. In describing the experiment, we followed the generally accepted practices commonly used in chemical and materials science journals. However, we understand that providing a more detailed and accessible description of the synthesis process would help broaden the readership. Therefore, we have included comprehensive information on the procedures for preparing film-forming solutions in the Supplementary Materials section. Additionally, we have added further details about the film formation process directly within the manuscript.

Round 3

Reviewer 1 Report

Comments and Suggestions for Authors

The revised manuscript is acceptable – the decision on whether to publish it is left to the editor of the journal.